# Enzalutamide induces cytotoxicity in desmoplastic small round cell tumor independent of the androgen receptor
Justin W. Magrath[1], Ilon N. Goldberg[1], Danh D. Truong [2], Alifiani B. Hartono[1], Shruthi Sanjitha Sampath[1], Chandler E. Jackson[1], Anushka Ghosh[1], Derrick L. Cardin[1], Haitao Zhang[1], Joseph A. Ludwig [2] & Sean B. Lee [1] ✉

Desmoplastic Small Round Cell Tumor (DSRCT) is a rare, pediatric cancer caused by the EWSR1::WT1 fusion protein. DSRCT predominantly occurs in males, which comprise 80-90% of the patient population. While the reason for this male predominance remains unknown, one hypothesis is that the androgen receptor (AR) plays a critical role in DSRCT and elevated testosterone levels in males help drive tumor growth. Here, we demonstrate that AR is highly expressed in DSRCT relative to other fusion-driven sarcomas and that the AR antagonists enzalutamide and flutamide reduce DSRCT growth. However, despite these findings, which suggest an important role for AR in DSRCT, we show that DSRCT cell lines form xenografts in female mice at the same rate as male mice and AR depletion does not significantly alter DSRCT growth in vitro. Further, we find that AR antagonists reduce DSRCT growth in cells depleted of AR, establishing an AR-independent mechanism of action. These findings suggest that AR dependence is not the reason for male predominance in DSRCT and that AR-targeted therapies may provide therapeutic benefit primarily through an AR-independent mechanism that requires further elucidation.

Desmoplastic Small Round Cell Tumor (DSRCT) is a rare, pediatric cancer caused by the EWSR1::WT1 fusion protein, which alters transcription and drives oncogenesis[1,2]. DSRCT is a member of the small round cell sarcoma tumor type which includes other oncogenic fusion protein driven tumors such as Ewing sarcoma, CIC::DUX4, and BCOR::CCNB3[3]. DSRCT survival remains poor with a 5-year survival rate of 15–25% necessitating the development of novel therapeutic strategies[4,5]. DSRCT is remarkable for its overwhelming predominance in males, who account for 80–90% of the patient population[4,6]. This male predominance stands out among other pediatric fusion oncogene-driven cancers such as Ewing Sarcoma[7,8] and alveolar rhabdomyosarcoma[9] which both demonstrate only a slight male predilection (approximately 60%). The reason for DSRCT's male predominance remains unknown and has the potential to provide insight into DSRCT biology and lead to the development of novel therapies. One hypothesis for the DSRCT male prevalence is that elevated testosterone levels in males active the androgen receptor (AR) pathway and drive tumor growth[10]. This hypothesis could help explain DSRCT's male predominance and the timing of tumor development, most commonly occurring in late

adolescence to early adulthood, the same time as testosterone levels increase in males[6].

A role for AR in DSRCT was first suggested by Fine et al. who found positive immunohistochemistry staining for AR in 10 of 27 DSRCT patients and demonstrated that DHT treatment increased the proliferation of patient-derived DSRCT cells in vitro. They further showed that DHT-induced proliferation could be reduced with flutamide treatment[11]. Based on these promising findings, Fine et al. treated six patients with combined androgen blockage including the first-generation AR blocker bicalutamide followed by the gonadotropin-releasing hormone Lupron. While three patients progressed on therapy, two patients experienced a 3-month partial response and another patient had stable disease for 3 months, suggesting some therapeutic benefit to targeting AR in DSRCT[11]. More recently, Lamhamedi-Cherradi et al. found that 75% of DSRCT patients from a set of 60 tumors stain positive for AR localized to the nucleus[10]. They further demonstrated that treatment with enzalutamide or an AR-directed antisense oligonucleotide reduced cell proliferation in the JN-DSRCT-1 cell line and a DSRCT patient-derived xenograft model[10]. Because many FDA-approved AR targeting treatments

[1]Department of Pathology and Laboratory Medicine, Tulane University School of Medicine, 1430 Tulane Ave, New Orleans, LA, USA. [2]Sarcoma Medical Oncology Department, The University of Texas MD Anderson Cancer Center, Houston, TX 77030, USA. ✉e-mail: slee30@tulane.edu

have been established in prostate cancer, these findings suggest a novel targeted treatment strategy in DSRCT could be quickly tested and brought to the clinic. However, further work must prove AR's role in the observed DSRCT cell and tumor growth reductions.

As a rare pediatric cancer, DSRCT model systems including cell lines and patient-derived xenograft models are scarce. Further, DSRCT cell lines have been thought to seed xenograft tumors poorly with studies commonly using 5–10 million cells for one xenograft, making large xenograft studies challenging[12–15]. We recently established a novel DSRCT cancer stem cell (CSC) model that expresses higher levels of stemness markers and resists chemotherapy treatment[16]. We sought to use this novel model to enable the establishment of DSRCT xenografts with a lower number of cells. Surprisingly, both the DSRCT CSC model and normal adherent culture cells were able to consistently seed tumors in NSG mice with as few as 100 cells. Despite DSRCTs overwhelming male predominance, studies have historically used female mice to seed DSRCT xenografts[14,15]. We hypothesized that mouse sex may explain our ability to seed DSRCT xenografts with only 100 cells, leading us to investigate the role of the androgen receptor in DSRCT. We expand on previous work by demonstrating that AR is expressed in the three most used DSRCT cell lines and that androgen treatment leads to AR nuclear localization. We further discovered that AR interacts with EWSR1::WT1 fusion protein leading the two proteins to co-occupy a subset of regulatory genomic regions. Surprisingly, we found that despite AR's expression and interaction with EWSR1::WT1, AR is dispensable for DSRCT cell survival and growth. These findings suggest sustained AR dependence is not the reason for male predominance in DSRCT and that AR-targeted therapies may provide therapeutic benefits to patients in an AR-independent manner.

## Results
### DSRCT cells form tumors at low seeding density
The use of 5–10 million DSRCT cells to establish xenografts is a significant limitation to examining DSRCT therapeutics in vivo. Utilizing our recently established DSRCT CSC model that employs serum-free conditions to increase stem cell characteristics[16], we sought to test whether these novel in vitro culture conditions could enable xenograft seeding with fewer cells. The JN-DSRCT-1 and BER-DSRCT cell lines were cultured in adherent or sphere (CSC model) conditions for 7 days and harvested for xenograft seeding. Because our previous work demonstrated the ability of sphere and adherent cultured cells to seed xenograft tumors with $10^6$ cells, we used $10^5$ as our highest cell seeding density. Ten-fold dilutions between $10^5$ and $10^2$ cells were seeded in NSG mice ($n = 8$ per seeding density) and tumor growth was monitored. For both cell lines, tumor growth occurred in a cell number-dependent manner with tumors first appearing in mice seeded with $10^5$ cells, followed by $10^4$, $10^3$, and finally $10^2$ (Fig. 1A). For the JN-DSRCT-1 tumors seeded from $10^5$ cells, tumors derived from spheres grew more quickly than those from adherent cells ($p = 0.0004$, Fig. 1A, B). However, this trend was not observed for tumors derived from BER-DSRCT cells or tumors derived from JN-DSRCT-1 at other cell seeding densities (Fig. 1A, B). Surprisingly, tumors formed in 8/8 mice for all JN-DSRCT-1 injections including when only 100 cells were injected (Fig. 1B). In BER-DSRCT, tumors formed in 8/8 mice for all injections of $10^5$ to $10^3$ cells, in 7/8 mice for $10^2$ BER-DSRCT cells derived from spheres, and in 6/8 mice for $10^2$ BER-DSRCT cells derived from adherent culture (Fig. 1B). We observed a greater variation in tumor size in groups originating from $10^2$ cells, which may be the result of small variations in cell seeding or growth coupled with a longer growth period (over 100 days). The third replicate in the BER-DSRCT $10^2$ adherent group was the only tumor found to invade the peritoneum and was by far the largest tumor in its group. As the peritoneum is the natural location of DSRCT tumors, this observation may reflect the influence of the microenvironment on tumor growth. Tumors derived from sphere and adherent cells as well as tumors derived from different seeding densities all displayed DSRCT morphology on H&E staining with small round blue cells and areas of desmoplasia (Fig. 1C, Supplementary Fig. 1). All tumors also stained positive for the proliferation marker KI67 (Fig. 1C, Supplementary Fig. 1).

The finding that both adherent and sphere culture cells are able to consistently seed tumors with as few as 100 cells was unexpected and a departure from the use of 5, 10, and as high as 50 million DSRCT cells to seed tumors in previous studies[12]. While the use of exceedingly large number of cells in these previous studies could be explained by following precedent, we created a table to compare the conditions used for tumor seeding in previous studies to see if there are other protocol differences that may explain our observation. While recent studies that seeded DSRCT xenografts were similar to our protocol in utilizing NSG mice and mixing cells with Matrigel, several studies used female rather than male mice (Supplementary Table 4). Given DSRCT's strong male predominance, we hypothesized that DSRCT xenograft tumor seeding ability may differ between male and female mice, potentially due to the influence of androgens. We therefore decided to investigate the role of AR in DSRCT.

### DSRCT cells express the Androgen Receptor
As an initial evaluation of the potential importance of AR in DSRCT, we used a set of gene expression data from fusion-positive sarcomas to compare the expression of AR in DSRCT ($n = 28$) to alveolar rhabdomyosarcoma (ARMS; $n = 23$), alveolar soft part sarcoma (ASPS; $n = 12$), Ewing sarcoma (ES; $n = 28$), and synovial sarcoma (SS; $n = 46$)[17]. DSRCT tumors express significantly higher levels of AR than the four other sarcoma types (Fig. 2A). Lamhamedi-Cherradi et al. previously demonstrated expression of AR in the JN-DSRCT-1 cell line[10]. We confirmed this finding and showed that AR is also expressed in the BER-DSRCT and BOD-DSRCT cell lines (Fig. 2B). AR protein expression is highest in BER-DSRCT, followed by JN-DSRCT-1, and the lowest in BOD-DSRCT. It is intriguing to note that robust AR expression is observed in BER-DSRCT, which is derived from tumors from a female patient[14]. To test the functionality of AR in DSRCT, immuno-fluorescence was used to examine nuclear localization of AR in cells cultured for 24 h with or without the androgen R1881 (1 nM). R1881 addition led to AR nuclear localization in all three DSRCT cell lines (Fig. 2C). To extend the findings of Lamhamedi-Cherradi et al. on the ability of AR blockers to reduce DSRCT viability, we treated DSRCT cell lines with vehicle control, 10 μM of enzalutamide, 10 μM darolutamide, or 10 μM flutamide with or without R1881 for 72 h. All three cell lines showed reductions in cell viability following AR blocker treatment (Fig. 2D, Supplementary Fig. 2A). The largest effect was seen for BER-DSRCT, with greater than 60% viability reduction observed, while JN-DSRCT-1 and BOD-DSRCT cells displayed modest reduction at less than 20%. Treatment with flutamide and dar-olutamide, but not enzalutamide, led to a significant reduction in viability for JN-DSRCT-1 and BOD-DSRCT. To assess the long-term effects of AR blockers, we performed 14-day colony formation assays where 1 or 10 μM of Enzalutamide or flutamide was added every 72 h. Enzalutamide and flutamide reduced colony formation in JN-DSRCT-1 and BER-DSRCT in a dose dependent manner with or without R1881 addition (Fig. 2E). Consistent with the 72-h treatment results, BER-DSRCT showed larger reductions in viability than JN-DSRCT-1. Notably, neither flutamide nor enzalutamide treatment reduced colony formation in BOD-DSRCT (Supplementary Fig. 2B). Immunofluorescence demonstrated that flutamide addition was able to reduce AR nuclear localization caused by R1881 treatment in all three cell lines (Fig. 2F, Supplementary Fig. 2C). The relative sensitivity of these three DSRCT cell lines to AR blockers aligns with the relative expression of AR in these cell lines: BER-DSRCT showed the highest expression of AR and was the most sensitive to AR blockers while BOD-DSRCT showed the lowest expression of AR and was the least sensitive to AR blockers.

### EWSR1::WT1 interacts with AR
Having shown that AR blockers reduce DSRCT viability, we next sought to better understand the role of AR in DSRCT. One potential explanation for the importance of AR could be that it acts as a critical downstream effector of the EWSR1::WT1 fusion protein. Utilizing previously established doxycy-cline (dox)-inducible cell lines that knockdown the expression of EWSR1::WT1[18], we examined the role of the fusion protein in AR

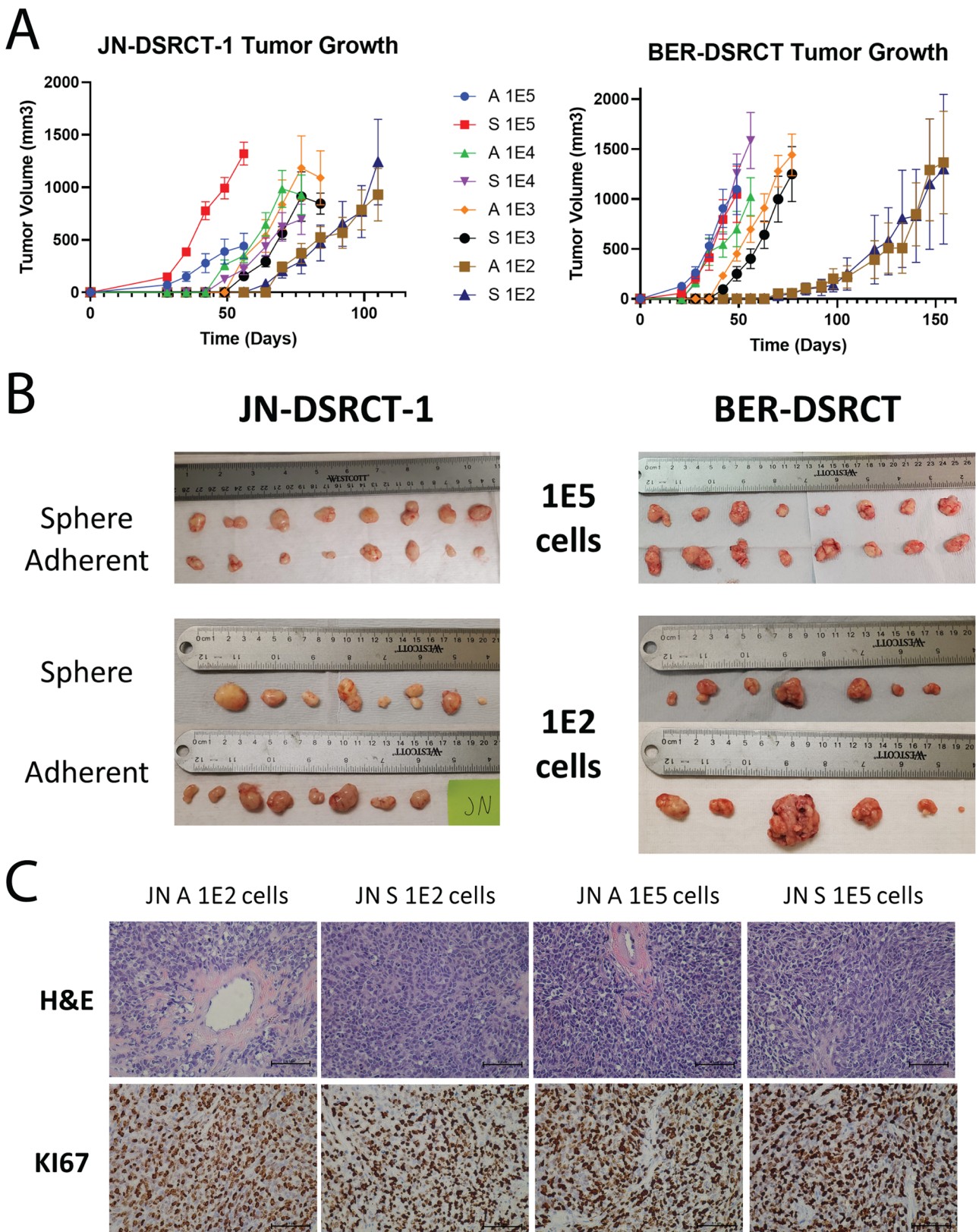

**Fig. 1 | DSRCT xenograft formation. A** Growth of DSRCT xenografts ($n = 8$ independent animals) seeded with $10^2$ to $10^5$ cells derived from (**A**) adherent or (**S**) sphere culture (Error bars = STD). **B** Final tumors derived from $10^5$ or $10^2$ DSRCT cells originally grown in sphere or adherent culture. Note: The largest tumor in the BER adherent $10^2$ group was found invading the peritoneum, likely explaining its enlarged size. **C** Representative H&E and KI67 staining of JN-DSRCT-1 xenografts seeded from (**A**) adherent or (**S**) sphere culture cells (scale bar = 50 μm).

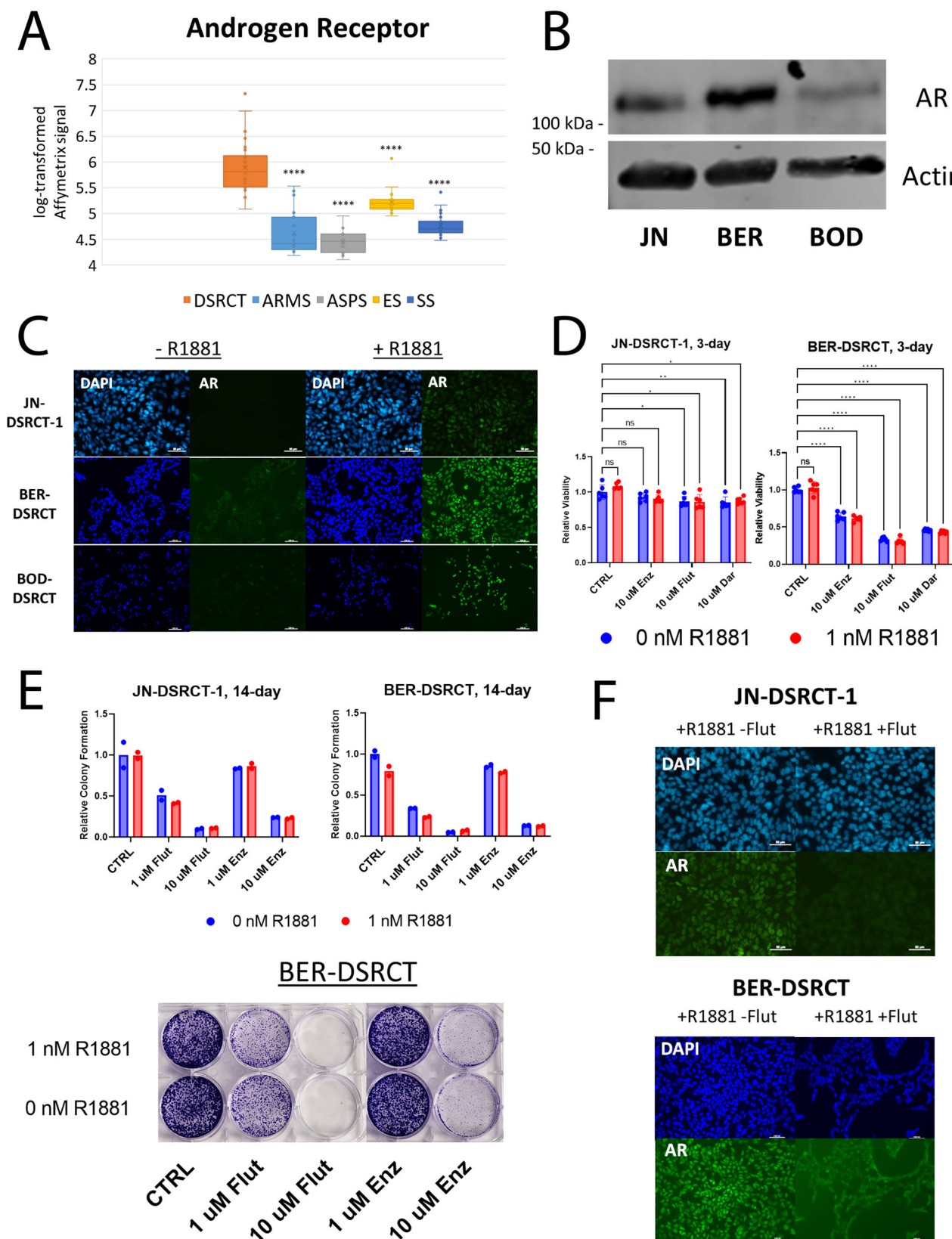

expression. EWSR1::WT1 knockdown led to reductions in AR at the protein level (Supplementary Fig. 3A) with JN-DSRCT-1 experiencing greater depletion of AR upon EWSR1::WT1 knockdown than BER-DSRCT. However, RT-qPCR found no significant difference in *AR* mRNA expression upon EWSR1::WT1 knockdown in JN-DSRCT-1 and found an

increase in *AR* expression in BER-DSRCT (Supplementary Fig. 3B). These results suggest EWSR1::WT1 does not regulate AR expression at the transcriptional level.

Another potential explanation for the importance of AR in DSRCT is that it may interact with the EWSR1::WT1 fusion protein and alter

**Fig. 2 | Androgen antagonists reduce DSRCT growth. A** Relative transcript levels of AR in DSRCT (*n* = 28) ARMS (*n* = 23), ASPS (*n* = 12), ES (*n* = 28), and SS (*n* = 46) primary tumors based on Affymetrix U133A expression array data (****$p < 0.0001$). **B** Western blot of AR protein expression in JN-DSRCT-1, BER-DSRCT, and BOD-DSRCT (representative from *n* = 3). **C** Immunofluorescence imaging of DAPI and AR in DSRCT cells treated with (+) or without (−) 1 nM R1881 for 24 h (*n* = 2 independent samples, scale bar = 50 μm). **D** Relative viability of JN-DSRCT-1 and BER-DSRCT cells treated with 10 μM enzalutamide, flutamide, or darolutamide for 72 h with or without 1 nM R1881 (*n* = 3 independent samples,

*$p < 0.05$, **$p < 0.01$, ***$p < 0.001$, ****$p < 0.0001$, Error bars = STD). **E** Colony formation assays examining the effect of flutamide and enzalutamide on DSRCT growth over a 14-day period (*n* = 2 independent samples) including quantification and a representative image for BER-DSRCT. **F** Immunofluorescence imaging of DAPI and AR in DSRCT cells treated with 1 nM R1881 and with (+) or without (−) 10 μM flutamide for 24 h (*n* = 2 independent samples, scale bar = 50 μm). Note: brightness was increased in **C** and **F** to improve visibility. Original images are available in Supplementary Fig. 2.

transcription. Lamhamedi-Cherradi et al. performed ChIP-seq to identify sites bound by AR in DSRCT and found the WT1 motif as one of the most enriched motifs bound by AR. Due to a lack of available C-term WT1 antibody, identifying direct binding sites of EWSR1::WT1 has been challenging. However, one set of ChIP-seq data using a discontinued C-term WT1 antibody in JN-DSRCT-1 cells is available[19]. Comparing the sets of AR-bound peaks and WT1-bound peaks, we identified 338 common peaks bound by both EWSR1::WT1 and AR (Fig. 3A). These sites could be regions where both EWSR1::WT1 and AR directly bind DNA. Alternatively, they may represent regions where only EWSR1::WT1 or AR directly binds DNA but both proteins are recruited to these regions due to the two proteins forming a complex. Homer motif enrichment analysis on the 338 commonly bound sites identified WT1 as the most enriched binding motif, while the AR binding motif was not significantly enriched among the peak set (Fig. 3A). To further delineate these two possibilities, we examined publicly available AR ChIP-seq data in a prostate cancer cell line. Only 10 of the identified 338 common peaks are bound by AR in LNCaP cells. This may be due to different chromatin states between DSRCT and prostate cancer cells, or could suggest a model in which AR binds to EWSR1::WT1, resulting in AR recruitment to WT1 binding sites (Fig. 3A). Genes co-occupied by AR and EWSR1::WT1 included previously identified EWSR1::WT1 regulated targets *MERTK*, *CCL25*, and *FGFR4* (Fig. 3B)[20,21].

We performed immunoprecipitation on nuclear extracts from BER-DSRCT and JN-DSRCT-1 with an antibody against AR to evaluate a potential interaction between AR and EWSR1::WT1. Pull down with AR antibody but not IgG control led to enrichment of AR and EWSR1::WT1 providing evidence of an interaction (Fig. 3C). AR pulldown not only enriched for EWSR1::WT1 but also wildtype EWSR1, suggesting the EWSR1 domain of EWSR1::WT1 as the likely location of interaction with AR. To further test the role of EWSR1 and WT1 domains in the fusion protein's interaction with AR, FLAG-tagged EWSR1 or WT1 were expressed in BER-DSRCT cells and immunoprecipitation with the AR antibody was performed. We observed a complex formation between AR and EWSR1 but not with WT1, providing further evidence that EWSR1 domains but not WT1 domains are involved in the interaction between AR and EWSR1::WT1 (Fig. 3D). Additionally, the reverse immunoprecipitation was performed in BER-DSRCT and JN-DSRCT-1 using an N-term EWSR1 antibody. Pull down with anti-EWSR1 resulted in enrichment of EWSR1, EWSR1::WT1, and AR (Supplementary Fig. 3C).

To gain insights into the genes regulated by the sites co-bound by EWSR1::WT1 and AR, we identified the closest gene to each peak and examined previously published RNA-seq data of JN-DSRCT-1 cells with or without siRNA knockdown of the EWSR1::WT1 fusion[21]. Of the genes co-occupied by AR and EWSR1::WT1, 89 (38.9%) are upregulated by EWSR1::WT1 in JN-DSRCT-1 and 31 (13.5%) are downregulated (Fig. 3E). Genes differentially expressed upon EWSR1::WT1 knockdown were more likely to have their co-occupied site in the promoter region and less likely to have the co-occupied site in the distal intergenic region (Fig. 3F). A scatterplot examining expression of the co-occupied genes differentially expressed by EWSR1::WT1 knockdown demonstrates concordance between their regulation in JN-DSRCT-1 and BER-DSRCT (Supplementary Fig. 3D). RT-qPCR was performed on cells treated with vehicle, 1 nM DHT, 10 μM enzalutamide, or both to determine if AR blockade could also alter expression of a panel of these co-occupied, EWSR1::WT1 regulated targets (*MERTK*, *FGFR4*, *EPHB3*, *CCL25*). Enzalutamide treatment with or

without the addition of DHT led to reductions in all 4 genes in JN-DSRCT-1 and 2/4 genes (*MERTK* and *CCL25*) in BER-DSRCT (Fig. 3G).

## DSRCT cells form tumors in male and female mice

Having expanded previous work on the role of AR in DSRCT to the BER-DSRCT and BOD-DSRCT cell lines and identified a previously uncharacterized interaction between EWSR1::WT1 and AR, we next examined if the importance of AR could explain our ability to seed xenografts with few DSRCT cells in male mice. JN-DSRCT-1 and BER-DSRCT cells grown in adherent culture were injected in 5 male and 5 female mice at $10^3$ cells per injection (10,000-fold lower than any published DSRCT xenograft seeding in female mice). Surprisingly, tumors formed in 5/5 injections in male and female mice for both cell lines (Fig. 4A, B). In JN-DSRCT-1 the tumors from female mice were larger than those in male mice, while for BER-DSRCT there was no statistical difference (Fig. 4A, B). Serum testosterone levels in male mice were significantly higher than in female mice as expected (Supplementary Fig. 4). AR expression and nuclear localization was observed in tumor cells formed in male but not in female mice for both JN-DSRCT-1 and BER-DSRCT (Fig. 4C). Despite the difference in AR, tumors in female mice showed DSRCT morphology and a high number of KI67-positive cells (Fig. 4C).

## DSRCT cell growth is independent of androgen receptor expression

The ability of DSRCT cells to form tumors in female mice with castrate levels of testosterone suggests androgens are not necessary for DSRCT growth and led us to question the importance of AR in DSRCT. Consistent with these findings, our previous colony formation and cell viability assays showed minimal growth stimulation with the addition of 1 nM of R1881 (Fig. 3, Supplementary Fig. 3). To evaluate the impact of androgens more comprehensively on DSRCT growth, we examined growth of JN-DSRCT-1 and BER-DSRCT cells after addition of dihydrotestosterone (DHT) (0.1 nM to 10 μM) or R1881 (0.01 nM to 1 μM). Neither cell line showed significant increase in growth upon addition of these androgens for 72 h (Fig. 5A, B). If the impact of androgens on DSRCT growth is only modest, we reasoned that 72 h may be insufficient to detect growth differences and normal FBS concentrations (even if charcoal stripped) may provide growth stimulation that obscures the effect of androgen. Therefore, we expanded our experiment to 12 days with R1881 concentrations between 0.1 nM and 10 nM and either 1, 5, or 10% charcoal stripped FBS. While JN-DSRCT-1 cells were unable to proliferate in 1% FBS, proliferation was observed in 5 and 10% FBS conditions and androgen addition led to significant increases in growth (Fig. 5C), consistent with previous findings[10]. Higher growth stimulation was observed in 5% FBS culture conditions (170% growth increase) than 10% FBS culture conditions (70% growth increase). In contrast, we did not observe an increase in growth with R1881 stimulation in BER-DSRCT cells grown at 10% FBS concentration, and BER-DSRCT cells failed to proliferate at FBS concentrations of 1% or 5% with or without R1881.

AR variants in prostate cancer can cause castration resistance and a similar mechanism could reconcile the importance of AR in DSRCT with our inconsistent findings of androgen growth stimulation[22–24]. However, Western blot analysis was unable to detect consistent expression of AR variants (Supplementary Fig. 5). To further evaluate the role of AR in DSRCT, we established cell lines utilizing a dox-inducible shRNA system to knockdown AR expression with four independent shRNAs, two targeting

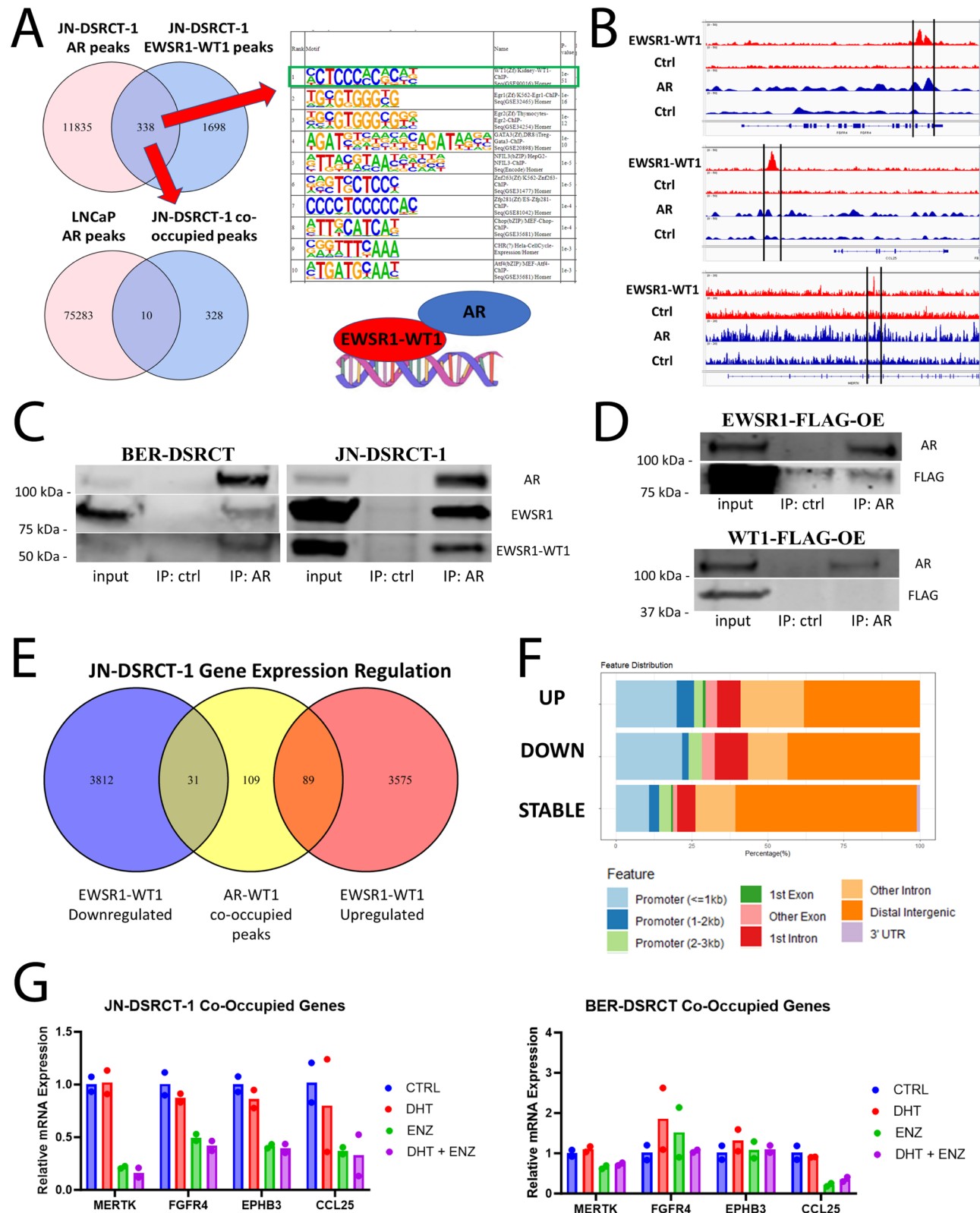

the AR N-terminus and two targeting the C-terminus[25]. RT-qPCR and Western blot confirmed successful AR knockdown in BER-DSRCT and JN-DSRCT cells with the addition of dox (Fig. 5D, E). All four shRNAs reduced AR expression, including an almost complete abrogation with shRNAs #1 and 3. It should be noted that the two N-terminus targeting AR shRNAs (#3 and 4) should deplete full length AR and its variants.

Despite near complete depletion of AR, DSRCT cells grew and formed colonies over a 14-day period (Fig. 5F, G). In JN-DSRCT-1, shAR #1 led to an increase in growth, while shAR #2–4 resulted in no significant growth difference. In BER-DSRCT, shAR #1 increased growth, shAR #2 showed no growth difference, while shAR #3 and #4 led to decreases in cell growth of 20% and 40%, respectively. Intriguingly, the BER shAR

**Fig. 3 | AR interacts with EWSR1::WT1. A** Venn diagram of binding site overlaps between AR and EWSR1::WT1 in JN-DSRCT-1 cells identifies a set of 338 co-occupied sites. HOMER motif analysis on these co-occupied sites identified the WT1 motif as the most highly enriched motif and failed to identify the AR binding site as enriched. Venn diagram comparing DSRCT co-occupied sites with AR binding sites in the prostate cell line LNCaP identifies only 10 common binding sites. **B** DNA tracks of ChIP-seq data of EWSR1::WT1, AR, or input controls in JN-DSRCT-1 demonstrating co-occupied sites in FGFR4, CCL25, and MERTK. **C** Immunoprecipitation-Western blot using anti-AR or control antibody demonstrating an interaction between AR and EWSR1::WT1 as well as native EWSR1 in JN-DSRCT-1 and BER-DSRCT cells. **D** Immunoprecipitation-Western blot using

anti-AR or control antibody in BER-DSRCT cells overexpressing FLAG tagged WT1 or EWSR1. **E** Venn diagram comparing the regulation of genes by EWSR1::WT1 (from RNA-seq data) and the closest gene to each genomic region co-occupied by EWSR1::WT1 and AR. **F** Genomic feature annotations of co-occupied sites that are associated with EWSR1::WT1 upregulated genes (UP), EWSR1::WT1 down-regulated genes (DOWN) or genes not significantly altered by EWSR1::WT1 (STABLE). **G** RT-qPCR of four EWSR1::WT1 upregulated and co-occupied genes (*MERTK, FGFR4, EPHB3, CCL25*) in DSRCT cells treated with vehicle ctrl (CTRL), 1 nM DHT (DHT), 10 μM enzalutamide (ENZ), or 1 nM DHT and 10 μM enzalutamide (DHT + ENZ) ($n = 2$ independent samples).

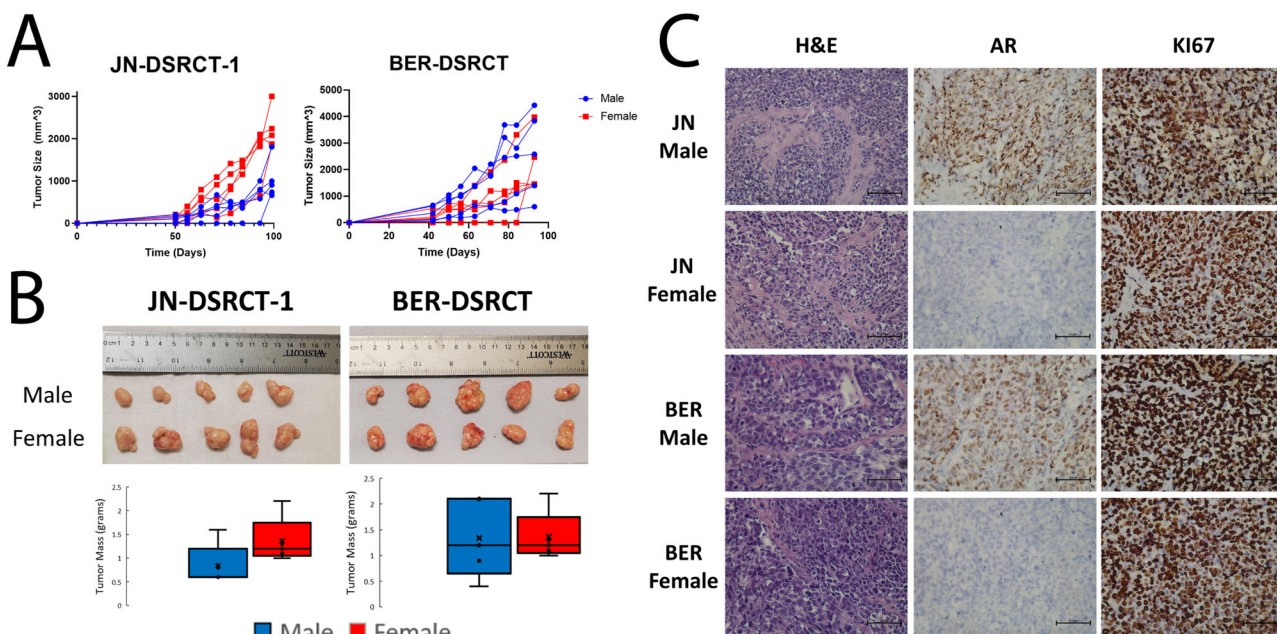

**Fig. 4 | DSRCT cells form tumors in male and female mice. A** Tumor growth of JN-DSRCT-1 and BER-DSRCT cells in male and female mice seeded with $10^3$ cells ($n = 5$ independent animals). **B** Images and mass of tumors derived from male and female mice ($n = 5$ independent animals, $*p < 0.05$). **C** Representative H&E, KI67, and AR staining of JN-DSRCT-1 and BER-DSRCT xenografts in male and female mice ($n = 3$, scale bar = 50 μm).

#4 led to the poorest knockdown of AR, despite being the shRNA that led to the greatest growth reduction.

**Enzalutamide reduces DSRCT growth independent of AR**

The finding that DSRCT cells can proliferate with near complete knockdown of AR suggests enzalutamide and flutamide may reduce DSRCT growth via a mechanism independent of AR. Colony formation assays with AR depleted using our shAR cell lines, confirmed this hypothesis by demonstrating that enzalutamide treatment reduces DSRCT growth even when AR expression is depleted by the addition of dox (Fig. 6A). As a second independent test, we measured cell viability on DSRCT cells treated with flutamide or enzalutamide for 72 h with or without dox-induced AR knockdown. For both flutamide and enzalutamide, dose response curves were nearly identical regardless of AR status (Fig. 6B). Previously we found (Fig. 3G) that enzalutamide treatment reduced the expression of several genes regulated by EWSR1::WT1 and co-occupied by AR and EWSR-WT1 (*MERTK, FGFR4, EPHB3, CCL25*). In contrast, RT-qPCR in JN- and BER-DSRCT shAR #1 and 3 cell lines did not show reduced expression in any of these genes with AR knockdown, suggesting these gene expression alterations are also caused by enzalutamide in an AR-independent manner (Supplementary Fig. 6A–D). One possible explanation for the ability of enzalutamide to reduce viability in DSRCT cells and alter expression of EWSR1::WT1 regulated genes is that enzalutamide is able to reduce EWSR1::WT1 expression, which has previously been shown to be critical to DSRCT growth[2]. Western blot of JN- and BER-DSRCT cells treated with

increasing doses of enzalutamide showed that high doses of enzalutamide treatment reduced expression of EWSR1::WT1 as well as downstream targets MERTK and LCK (Fig. 6C). In contrast, AR depletion by shAR #1 and 3 did not lead to reduced protein expression of EWSR1::WT1, MERTK, or LCK (Supplementary Fig. 6E). RT-qPCR found that EWSR1::WT1 expression is reduced at the transcriptional level with 10 μM enzalutamide treatment with or without the addition of 1 nM DHT (Fig. 6D). Together these findings suggest enzalutamide cytotoxicity in DSRCT is independent of AR and may involve downregulation of the EWSR1::WT1 oncogene.

This AR-independent cytotoxicity may occur through another member of the NR3 family of nuclear receptors. Analysis of RNA-seq data from 22 DSRCT tumors[26] found expression of a diverse range of NR3 family members including *AR*, glucocorticoid receptor (*NR3C1*), estrogen receptor (*ESR1*), and progesterone receptor (*PGR*) (Fig. 6E). Intriguingly, we observed an inverse correlation between *AR* expression and expression of *NR3C1*, *ESR1*, and *PGR*, with higher expression of the later three nuclear receptors observed in tumors with lower AR expression. Bulk RNA-seq data from three DSRCT patient-derived xenografts (PDXs) similarly showed expression of a diverse range of NR3 family members including expression of *NR3C1* (3/3 tumors), *AR* (2/3 tumors) and *ESR1* (2/3 tumors) (Supplementary Fig. 7A). Expression of *NR3C1* and *AR* was higher in DSRCT PDXs as compared to Ewing sarcoma and CIC-DUX4 PDXs. Single nucleus RNA-seq (snRNA-seq) of the same three DSRCT PDXs enabled the examination of intra- and inter-tumoral heterogeneity. Within each PDX, we see subsets of DSRCT cells with different expression profiles of *AR*, *NR3C1*, and *ESR1*

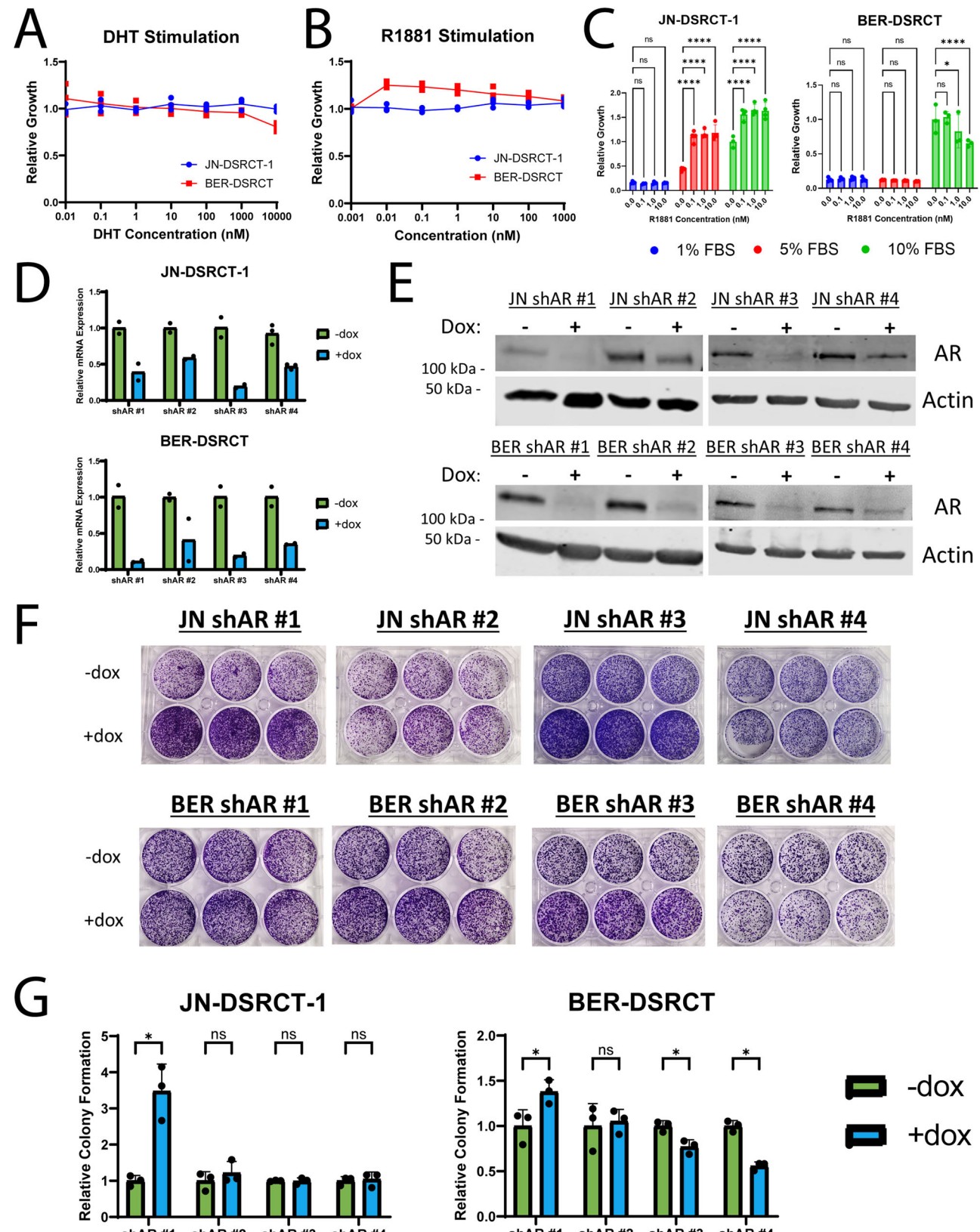

**Fig. 5 | DSRCT growth is independent of AR expression. A** Relative growth of JN-DSRCT-1 (blue) and BER-DSRCT (red) cells treated for 72 h with DHT (0.1 nM to 10 μM, $n = 3$ independent samples). **B** Relative growth of JN-DSRCT-1 (blue) and BER-DSRCT (red) cells treated for 72 h with R1881 (0.01 nM to 1 μM, $n = 3$ independent samples). **C** Relative growth of JN-DSRCT-1 and BER-DSRCT cells cultured in 1%, 5%, or 10% FBS and treated for 12 days with R1881 concentrations between 0.1 and 10 nM ($n = 4$ independent samples, *$p < 0.05$, ****$p < 0.0001$,

Error bars = STD). **D** RT-qPCR of AR expression in JN-DSRCT-1 and BER-DSRCT shAR #1–4 cell lines with or without dox ($n = 2$–3 independent samples). **E** Western blot of AR protein expression in shAR #1–4 cell lines with or without dox (representative blot, $n = 2$ independent samples). **F** Images of colony formation assays from shAR #1–4 cell lines ($n = 3$ independent samples). **G** Quantification of colony formation assays of shAR #1–4 cell lines with (blue) or without dox (green) ($n = 3$ independent samples, *$p < 0.05$, **$p < 0.01$, Error bars = STD).

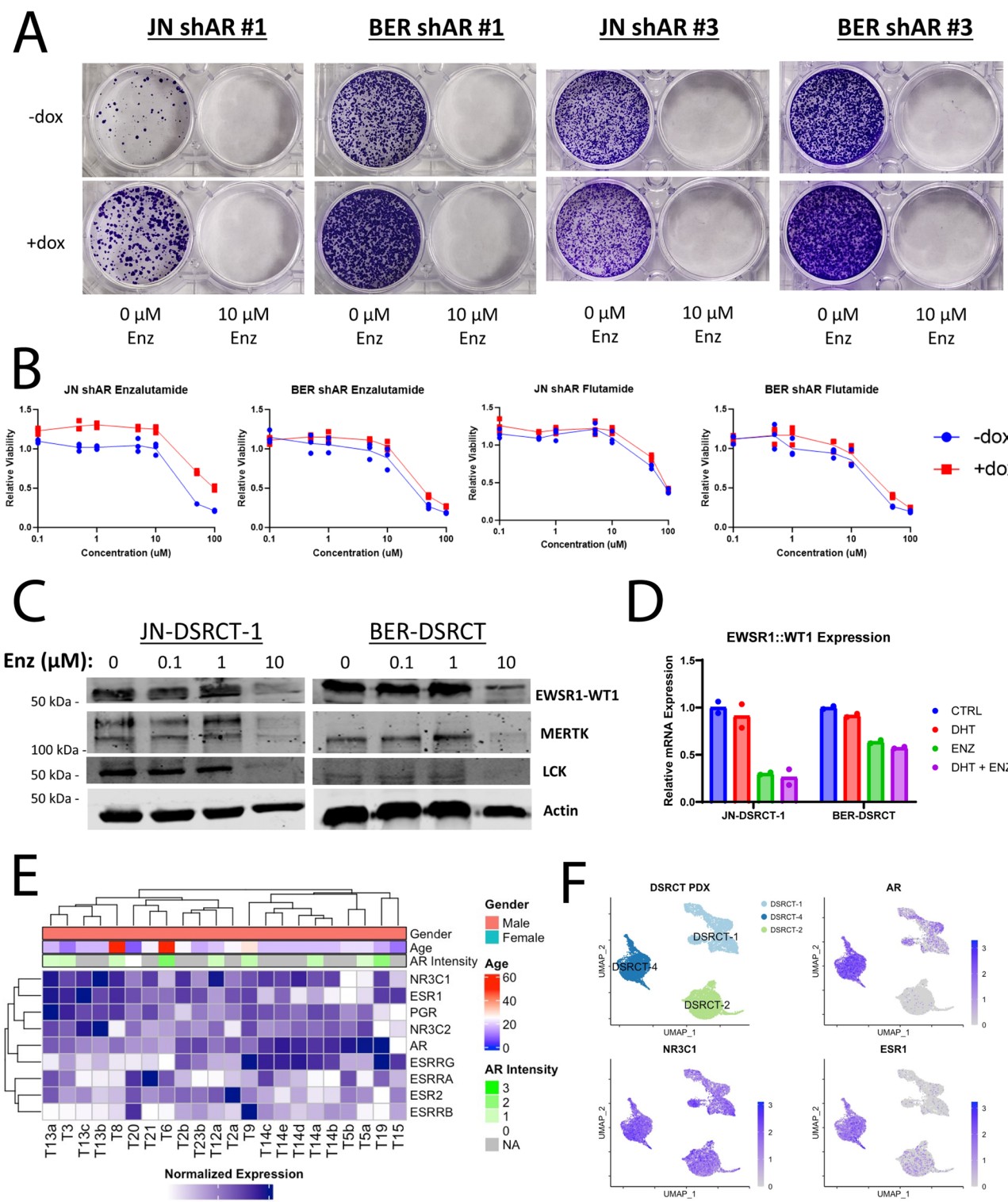

**Fig. 6 | Enzalutamide reduces DSRCT growth independent of AR. A** Colony formation assays of JN-DSRCT-1 and BER-DSRCT shAR #1 and 3 cell lines treated for 14-days with vehicle or 10 µM of enzalutamide in conjunction with dox (+) addition to deplete AR ($n = 2$–3 independent samples). **B** Relative viability of JN-DSRCT-1 and BER-DSRCT shAR #3 cell lines pretreated with or without dox to deplete AR and then treated for 72 h with concentrations of enzalutamide or flutamide ($n = 3$ independent samples). **C** Western blot examining EWSR1::WT1, MERTK, LCK, and ACTIN of DSRCT cells treated with 0, 0.1, 1, or 10 µM enzalutamide for 72 h. Proteins of similar sizes were detected on the same membrane via chemiluminescent detection. **D** RT-qPCR of DSRCT cells treated for 72 h with vehicle ctrl (CTRL), 1 nM DHT (DHT), 10 µM enzalutamide (ENZ), or 1 nM DHT and 10 µM enzalutamide (DHT + ENZ) ($n = 2$ independent samples). **E** Heatmap of NR3 nuclear receptor expression in DSRCT tumors ($n = 22$). AR intensity was measured from matched IHC. **F** UMAP projections showing expression of AR, NR3C1, and ESR1 in snRNA-seq from three DSRCT patient-derived xenografts.

(Fig. 6F, Supplementary Fig. 7B). For instance, *NR3C1* expression was observed in most cells from all three PDXs, but the *AR* and *ESR1* expression was seen in mainly cells from DSRCT-4 with only a small fraction of cells from DSRCT-1 and DSRCT-2 expressing *AR* and *ESR1*. Cells expressing *AR*, *ESR1*, and *NR3C1* were co-expressed with epithelial markers (*CDH1*, *KRT23*) while cells expressing only *NR3C1* lacked epithelial markers and instead expressed mesenchymal (*CDH2*, *DES*) and neuronal (*NCAM*) markers (Supplementary Fig. 7B). Together, these data demonstrate the heterogeneity of NR3 nuclear receptor expression in DSRCT and identify an intriguing future area for investigation.

## Discussion

In this work, we investigate and provide clarity on two outstanding questions in the DSRCT field: the number of cells needed to seed xenograft tumors and the importance of the androgen receptor. The first publication to establish xenograft tumors from a DSRCT cell line used 50 million JN-DSRCT-1 cells in SCID mice[12]. More recently 5–10 million cells have been utilized to establish xenografts in NSG mice[14,15]. However, the prevailing wisdom has remained that establishing xenografts from DSRCT cell lines is challenging and a barrier to the investigation of potential therapeutics. Using our novel in vitro DSRCT CSC model, we sought to create a new protocol to enable the establishment of DSRCT xenografts with fewer cells. To our great surprise, we were able to seed xenograft tumors with as few as 100 cells from both DSRCT CSC culture and standard adherent culture. We hypothesized that mouse sex may contribute to the ability to establish xenograft tumors with so few cells. However, further investigation showed that tumors could be seeded from $10^3$ adherent cells in either male or female mice. Advances in xenograft tumor seeding protocols including the use of Matrigel and more severely immunocompromised mice (NSG versus SCID) have enabled seeding xenografts with fewer cells in multiple cancer types[27–29]. It is possible that 5–10 million DSRCT cells have continued to be used when these advances have no longer required it. This study is the first comprehensive evaluation of DSRCT xenograft tumor seeding and demonstrates that DSRCT xenografts can be reliably formed with far fewer cells than previously utilized. This new understanding will enable the testing of DSRCT therapeutics in vivo with fewer required cells and help to accelerate the testing of novel therapies.

With a 5-year survival rate of 15–25% and a high frequency of recurrence and metastasis, DSRCT is in urgent need of novel therapeutic strategies to improve patient outcomes. The presentation of DSRCT with tens to hundreds of independent nodules makes complete surgical resection extremely challenging and the prioritization of systemic therapy of paramount importance. Lamhamedi-Cherradi et al. recently proposed AR as a novel therapeutic target in DSRCT and demonstrated the ability of the AR blocker enzalutamide and an AR-targeting anti-sense oligonucleotide to reduce DSRCT tumor growth in vivo[10]. Our hypothesis that mouse sex may play a role in DSRCT xenograft seeding ability led us to further investigate the role of AR in DSRCT. Consistent with the findings of Lamhamedi-Cherradi et al., we found expression of AR in JN-DSRCT-1 and two other DSRCT cell lines, AR nuclear localization in response to androgen, and reduced cell growth in response to enzalutamide treatment (Fig. 2). Our findings of DSRCT sensitivity at 10 µM are similar to findings by Wu et al. who showed an enzalutamide IC50 in the range of 5–30 µM[26], much higher than the findings of Lamhanedi-Cherradi et al. of an IC50 of 0.046 µM[10]. While all these experiments were performed on the same JN-DSRCT-1 cell line, differences in culture conditions, treatment length, and treatment frequency may help explain these discrepancies. Despite the sensitivity of DSRCT to high doses of enzalutamide and flutamide, we found that DSRCT cells can form tumors in female mice and proliferate without AR nuclear localization. While we did not observe increased proliferation with R1881 or DHT treatment in vitro at 72-h, we were able to observe R1881-induced growth stimulation in JN-DSRCT-1 cells after 12-days of culture. Growth stimulation was higher in conditions with less FBS, suggesting androgens can induce DSRCT proliferation in cells deprived of other mitogens. This may explain the discrepancy in our in vitro and in vivo findings. The existence of other growth stimulants in vivo may make androgen stimulation unnecessary in female xenograft mice.

Three explanations for the lack of importance of androgens in DSRCT growth in vivo are that (1) DSRCT cells rely on the androgen receptor in a ligand-independent manner, (2) DSRCT cells lines, having been cultured for prolonged periods without DHT supplementation, have become AR indifferent, or (3) AR is not necessary for DSRCT growth. AR splice variants have been shown to promote castration resistance in prostate cancer[22,23]. However, we were unable to identify splice variants with Western blot analysis in DSRCT. Similarly, Lamhamedi-Cherradi et al. did not find expression of AR-V7 by IHC in a set of 12 DSRCT tumors[10]. AR variants typically overcome castrate levels of androgens by localizing to the nucleus independent of androgen[22]. Our observation that androgens are necessary for nuclear localization of AR in DSRCT cell lines (Fig. 2C) thus further casts doubt on AR playing a castration-resistant role in DSRCT. By establishing four dox-inducible shAR cell lines that deplete AR, including two shRNAs targeting the N-terminus of AR (which would also deplete AR variants) and two C-terminus targeting shRNAs, we were able to determine the importance of AR in DSRCT cell lines. Despite near complete abrogation of AR expression, JN-DSRCT-1 and BER-DSRCT cells retained the ability to proliferate and form colonies, demonstrating AR is dispensable for DSRCT growth in vitro. In prostate cancer, an AR indifferent phenotype has been established featuring reduced sensitivity to AR blockers and hyper-activation of the E2F cell-cycle master regulator[30]. Given the transient clinical responses observed by Fine et al. in two out of six DSRCT patients receiving combined hormonal blockade[11], there may be a subpopulation of AR-responsive tumors not adequately modeled by current cell lines and paired xenografts. The creation and testing of new DSRCT PDX models, better reflecting DSRCT tumor biology including tumor-associated stromal cells, will be critical in further exploring the AR-independence we observe in DSRCT cell lines.

Our finding that AR is not necessary for DSRCT xenograft growth leads to questions about the target for enzalutamide in DSRCT cell lines and the reason behind DSRCT's overwhelming male predominance. While AR knockdown failed to reduce DSRCT cell growth, treatment with 10 µM of enzalutamide led to colony formation reductions of over 75% in JN-DSRCT-1 and BER-DSRCT. This suggests enzalutamide may target an alternative, AR-independent pathway critical to DSRCT survival. This alternative target could be another member of the NR3 family of nuclear receptors such as the glucocorticoid receptor or estrogen receptor. Our examination of NR3 family member expression in a set of 22 DSRCT patients and three PDXs found high expression of glucocorticoid receptor (*NR3C1*) and an intriguing inverse relationship between *AR* expression and the expression of *NR3C1*. Future work examining the function of NR3C1 in DSRCT and its relationship with AR may provide greater insight into enzalutamide's activity in DSRCT. Consistent with this hypothesis, Smith et al. identified three prostate cancer cell lines that lack AR expression but still were sensitive to enzalutamide treatment as a result of the glucocorticoid receptor[31]. Likewise, in breast cancer, cytotoxic effects of enzalutamide have been shown to be independent of AR and instead reliant on targeting the estrogen receptor[18]. Intriguingly, we found that treatment with 10 µM of enzalutamide reduced EWSR1::WT1 expression, which could explain its cytotoxicity. Further studies are necessary to determine if this decrease in EWSR1::WT1 is directly caused by enzalutamide and the effector pathway for enzalutamide induced cytotoxicity in DSRCT or whether fusion gene decrease is a byproduct of the enzalutamide-induced death.

If androgens and AR are not needed for DSRCT growth, then androgen growth dependence cannot explain the strong male prevalence in DSRCT. One potential alternative explanation is that androgen is necessary for the formation of the EWSR1::WT1 fusion but not subsequent tumor growth. This explanation could also help explain the absence of a difference in survival based on patient sex[6]. Lin et al. demonstrated that translocations in prostate cancer involving TMPRSS2, ERG, and ETV1 are non-random events caused by AR[17]. DHT stimulation triggers co-localization of TMPRSS2 and ERG genomic regions through AR binding which, when

exposed to radiation, triggers chromosomal breakage and non-random translocations[17]. A similar process could occur in the DSRCT cell of origin to initiate EWSR1::WT1 translocation. This hypothesis is bolstered by recent findings from Nicholas et al. showing that AR binds to introns 5 and 8 of EWSR1 in prostate cancer and further, that R1881 treatment can increase chromosomal breakage at this site in an R-loop dependent manner[32]. In fact, intron 7 of EWSR1, located between the two AR binding sites, is the most common breakpoint location in DSRCT. This concept is further supported by Lamhamedi-Cherradi et al.'s discovery that AR binds to the native WT1 genomic location in DSRCT[10]. Future testing of this hypothesis will not only improve understanding of DSRCT's male prevalence and the mechanism of EWSR1::WT1 formation but may also provide insight into the cell types that are conducive to such a translocation: the potential DSRCT cell of origin.

## Methods

### Cell lines and culture conditions
JN-DSRCT-1, BER-DSRCT, and BOD-DSRCT cell lines have been described previously and validated for the defining EWSR1::WT1 fusion[12,14,33]. Adherent culture: cells were grown on tissue culture (treated) plates in DMEM/F12 media without phenol red supplemented with 10% charcoal-stripped FBS (Gibco), 2mM L-Glutamine, 100 U/mL penicillin and 100 μg/mL streptomycin (ThermoFisher, Waltham, MA). Sphere culture: $4 \times 10^6$ cells were seeded on non-treated plates (Costar® 6-well Clear Not Treated Multiple, Corning) in a 1:1 mixture of DMEM/F12 and Neurobasal Media supplemented with 100 U/mL penicillin, 100 μg/mL streptomycin, and 2 mM L-Glutamine (ThermoFisher, Waltham, MA). Media were changed every two days.

### Xenografts in immune-deficient mice
Animal procedures were approved by the Tulane Institutional Animal Care and Use Committee (Protocol: 1500). Male or Female NOD-SCID-IL2Rγ-null (NSG) mice (6 weeks) were purchased (Jackson Laboratory, Bar Harbor, ME) and used for all xenograft studies. The maximum allowable tumor size was 15 mm in the longest diameter, which was not exceeded as measured from the mouse surface. This standard was also met for the largest BER-DSRCT tumor which predominantly grew intraperitonially making it not measurable on the mouse surface. We have complied with all relevant ethical regulations for animal use. JN-DSRCT-1 and BER-DSRCT cells grown in adherent culture or sphere culture for 7 days were counted, mixed in a 1:1 ratio of media to Matrigel (Corning, Tewksbury, MA), and 200 μL of cell mixture containing $10^5$, $10^4$, $10^3$, or $10^2$ cells was subcutaneously injected into the lower flanks of NSG mice with adherent cells injected in the left flank and sphere cells injected in the right flank. Tumor volume was measured weekly with calipers and calculated: length × (width)$^2$ × 0.5 (length is largest diameter, width is perpendicular to the length). Mice were sacrificed when tumor volume reached >2000 mm$^3$. Tumors were harvested and weighed. Tumor fragments were fixed in formalin for immunohistochemistry analysis. Fixed tissues were embedded in paraffin, sectioned (5 μm), stained with H&E or antibodies (AR, KI67), and imaged (Nikon Eclipse 80i microscope; NIS-Elements software, Melville, NY).

### Protein isolation and western blot analysis
Cell lysates were harvested with RIPA lysis buffer supplemented with 1 mM NaF, complete EDTA-Free Protease Inhibitor Cocktail (Sigma-Aldrich), and 2 mM Na$_3$VO$_4$. Proteins were resolved in 10% SDS-PAGE gels and transferred onto a 0.45 μm nitrocellulose membrane (Bio-Rad). Membranes were blocked with 5% non-fat milk and incubated with primary antibodies at 4 °C overnight, followed by secondary antibodies LI-COR IRDye 680RD goat anti-Rabbit (#926- 68071, 1:10,000 dilution) or LI-COR IRDye 800CW goat anti-Mouse (#926-32210, 1:10,000 dilution) and scanned on LI-COR Odyssey CLx (Lincoln, NE). At least two independent immunoblots were performed for each experiment, with a representative immunoblot shown. Antibodies are listed in Supplementary Table 2.

### AR immunofluorescence
Cells were seeded on coverslip culture plates (Lab-Tek Chamber Slide System, Thermo Scientific) and treated for 24 h with or without 1 nM R1881 and/or 10 μM of androgen receptor blockers. Slides were fixed with 4% paraformaldehyde, permeabilized with 0.5% Triton X, washed 3X with PBS, blocked with 5% BSA, washed with PBS, incubated with AR primary antibody for 1 h at room temperature, washed 3X with PBS, incubated with secondary antibody (Zenon Alexa Fluor 488 Rabbit, Invitrogen), washed 3X with PBS, and incubated with DAPI for nuclear staining. Fluorescence imaging was performed with Nikon Eclipse 80i microscope using NIS-Elements software.

### Colony formation assays
Cells were seeded at 10,000 cells per well in a 6-well plate. Media containing indicated treatment were changed every two days. After 14-days, plates were washed with PBS and stained with 0.5% crystal violet in 10% methanol for 20 minutes, followed by two washes with water and overnight drying. Stain was removed with 1 mL methanol and quantified by measuring absorbance at 570 nm using Clariostar microplate reader (BMG Labtech, Cary, NC).

### Co-Immunoprecipitation experiments
DSRCT cells in 10 cm plates were incubated on ice with hypotonic lysis buffer (10 mM Tris pH7.9, 1.5 mM MgCl$_2$, 10 mM KCl, 0.5 mM dithiothreitol (DTT), and 0.5 mM phenylmethylsuffonyl fluoride (PMSF) for 5 min. Cells were scraped and lysed by Dounce homogenization followed by centrifugation at $6000 \times g$ to isolate nuclei. Nuclear extracts were extracted at 4 °C in high salt buffer (20 mM Tris pH7.9, 1.5 mM MgCl$_2$, 25% glycerol, 0.2 mM EDTA, 0.5 mM DTT and 0.3 M NaCl) followed by centrifugation at $6000 \times g$. Nuclear lysates were split in two and incubated overnight with 2 μg of primary antibody (AR, EWSR1, or FLAG) or IgG control while rotating at 4 °C. Pre-cleared magnetic protein G beads were used to pulldown antibody-bound proteins, followed by three washes and elution in sample buffer. Pulldown products were subsequently analyzed by Western blot analysis.

### Bioinformatics analysis
ChIP-seq data of anti-AR and anti-WT1 antibodies in JN-DSRCT-1 cells (GSE151380, GSE156277), ChIP-seq data of anti-AR antibodies in LNCaP cells (GSE28264[34]), and RNA-seq data following EWSR1::WT1 knockdown in JN-DSRCT-1 and BER-DSRCT cells (GSE137561[21]) were retrieved from GEO database. Co-occupied peaks were identified with Bedtools[35] and annotations were established with ChIPseeker[36]. HOMER was used for motif enrichment analysis[37]. The Integrative Genomics Viewer (IGV) was used for visualization of peaks from BigWig files[38].

### RNA isolation and Real-Time qPCR analysis
Total RNA was isolated with RNA-STAT60 (Tel-Test, Friendswood, TX) and 750 ng of RNA was reverse transcribed to generate cDNAs using iScript cDNA Synthesis Kit (Bio-Rad, Hercules, CA). Relative transcript levels were analyzed by real-time qPCR using SYBR Green (SsoAdvanced Universal SYBR Green Supermix, Bio-Rad) and calculated by the comparative Ct method normalized against human ACTB (β-ACTIN) for cell culture. Primers are listed in Supplementary Table 1.

### Cell growth and viability assays
Cell growth and viability assays were both performed using CCK-8 assay (Sigma-Aldrich) per the manufacturer's directions. For cell growth assays, cells were seeded in 96-well plates in charcoal-stripped media on day 0. On day 1, doses of dihydrotestosterone (DHT) (0.1 nM to 10 μM) or R1881 (0.01 nM to 1 μM) were added to wells in triplicate. CCK-8 was performed 72 h or 12 days later. For viability assays cells were seeded in 96-well plates with or without doxycycline to knockdown AR. Two days later doses of enzalutamide and flutamide between 0.1 and 100μM were added. After 72 h, the CCK-8 assay was performed to assess viability. For all experiments,

absorbance was measured using Clariostar microplate reader (BMG Labtech, Cary, NC).

## Generation of dox-inducible shRNA cell lines

Doxycycline (dox)-inducible LT3-GEPIR vector[25] was used to generate stable cell lines in JN-DSRCT-1 and BER-DSRCT cells. Annealed oligonucleotides containing shRNA sequences against AR or the 3′UTR of WT1 were inserted into XhoI and EcoRI sites of the vector (Supplementary Table 3). Lentivirus was created by co-transfecting HEK293T cells with the LT3-GEPIR-shWT1 lentiviral vector and ViraPower lentiviral packaging mix (Invitrogen) using Lipofectamine3000 (Thermofisher Scientific). Viral supernatants were collected 48-, 72-, and 96 h post-transfection, and concentrated with LentiX-Concentrator (Takara Bio, San Jose, CA). JN-DSRCT-1 and BER-DSRCT cells were transduced with LT3-GEPIR-shWT1 in the presence of polybrene (10 µg/ml) for 16 h. Cells were selected with puromycin (0.5 µg/mL for BER-DSRCT, 2 µg/mL for SK-DSRCT2) at 48 h post-transduction. Stable cell lines were validated by RT-qPCR and Western blot analyses with or without dox.

## RNA-seq analysis of DSRCT patients

RNA-seq data of 14 DSRCT patients were acquired from the European Genome-phenome Archive under the study accession number: EGAS00001004575. In compliance with all relevant ethical guidelines, patients provided written informed consent to collect and use their tumor specimens for research purposes using lab protocols LAB08-0151 or LAB04-0890, which are approved by MDACC's Institutional Review Board. The charts and electronic medical records of patients with a confirmed diagnosis of DSRCT were included for analysis and archived at the MDACC biospecimen bank or Dr. Ludwig's laboratory. DSRCT specimens generated from patients treated at MDACC from 1990 to 2019 were used to create a TMA for analysis of AR staining[10]. Specialist pathologists used clinical information, immunohistochemistry, and cytogenic analysis for the fusions to confirm the diagnoses. Tissue collection and data processing were described previously[26]. Heatmaps were produced using the R package ComplexHeatmap[39].

## Collection of tissue from PDX

All experiments were conducted per protocols and conditions approved by the University of Texas MD Anderson Cancer Center (MDACC; Houston, TX) Institutional Animal Care and Use Committee (eACUF Protocols #00000712-RN02). Male NOD (SCID)-IL-2Rgnull mice (The Jackson Laboratory; Farmington, CT) were subcutaneously injected with PDX explants (2 mm) to generate xenografts. All mice were maintained under barrier conditions and treated using protocols approved by The University of Texas MD Anderson Cancer Center's Institutional Animal Care and Use Committee. The maximum allowable tumor size of 2000 mm³ was not exceeded. We have complied with all relevant ethical regulations for animal use. DSRCT, ES, and CDS PDX lines were generated from the Sarcoma Tissue Bank at MD Anderson Cancer Center and maintained by the Ludwig lab. In compliance with all relevant ethical guidelines, patients provided written informed consent to collect and use their tumor specimens for research purposes using lab protocols LAB08-0151, which are approved by the Institutional Review Board at MD Anderson Cancer Center. Once the tumors reached a volume of 2000 mm³, tumors were explanted, and a portion was flash-frozen for snRNA-seq and RNA-seq. All ethical regulations relevant to human research participants were followed.

## RNA-seq analysis of DSRCT PDX

Total RNAs of DSRCT PDX frozen samples were extracted using RNeasy Mini Kit (Qiagen) and libraries were made using the KAPA Stranded RNA-Seq Library Preparation Kit (KAPA Biosystems). Libraries from each sample were pooled together and sequenced on the Illumina HiSeq 4000 (Illumina). Reads were mapped to the hg19 reference genome using GSNAP[40]. For gene expression calculations, raw counts were obtained using featureCounts[41] and were then normalized by regularized log transformation in DESeq2[42]. Data was uploaded to GEO and can be accessed at GSE245914.

## snRNA-seq analysis of DSRCT PDX

snRNA-seq data from PDX lines were obtained from a previous study GSE200529[43]. We used Cell Ranger mkfastq to generate demultiplexed FASTQ files. Reads were mapped with both introns and exons in Cell Ranger 5.0 using the include-introns option for counting intronic reads[44]. Reads were aligned to the human GRCh38 genome, and reads were then quantified as UMIs by Cell Ranger count. We performed QC and normalization based on guidelines for QC from OSCA and others[45]. We inspected UMIs, gene counts, and the percentage of mitochondrial genes and identified outliers based on median absolute deviation (MAD). We used a strict value of 2 or more MADs from the median while also using generic cut-offs. Cells that did not meet the criteria were removed from the analysis. Scrublet was used to predict and detect doublets within the data[46]. Seurat v3 was used for sample normalization, dimensional reduction, scaling, and UMAP visualization[47].

## Statistics and reproducibility

Two or more independent replicates were utilized for experiments. Two-way ANOVA and the Student's $T$ test was used as appropriate using GraphPad Prism 7 program (GraphPad Software, San Diego, CA). A $p$-value < 0.05 was considered statistically significant. For western blots, IHC and immunofluorescence at least two replicates were performed, and the same trend was seen in all replicates.

## Reporting summary

Further information on research design is available in the Nature Portfolio Reporting Summary linked to this article.

## Data availability

RNA-Seq data of DSRCT PDXs was uploaded to GEO and can be accessed at GSE245914. Previously published RNA-seq data of 14 DSRCT patients were acquired from the European Genome-phenome Archive under the study accession number: EGAS00001004575. snRNA-seq data from PDX lines were obtained from a previous study available at GSE200529. Original western blots are available in Supplementary Figs. 8–15. Original data is available in Supplementary Data 1.

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

## Acknowledgements

This work was supported by the NCI R01CA222856 (S.B.L.) and the Tulane Carol Lavin Bernick Faculty Investment Fund (S.B.L.). The Advanced Technology Genomics Core (ATGC) is supported by the Core grant CA016672 (ATGC) and NIH 1S10OD024977-01. J.A.L. and D.D.T. are supported with generous philanthropic funds from the Cory Monzingo Foundation and Blake Abercrombie Foundation. D.D.T. would like to acknowledge funding support from Daniel Benedict Gazan Fellowship in Sarcoma Research. The authors acknowledge the support of the High-Performance Computing for research facility at the University of Texas MD Anderson Cancer Center for providing computational resources that have contributed to the research results reported in this paper. We also acknowledge the MD Anderson Advanced Cytometry and Sorting Facility for their assistance in isolating nuclei from DSRCT PDX tissue.

## Author contributions

J.W.M., S.B.L., D.L.C., H.Z., D.D.T., and J.A.L. designed experiments. J.W.M., I.N.G., D.D.T., A.B.H., S.S.S., C.E.J., and A.G. conducted experiments. J.W.M., I.N.G. and D.D.T. analyzed data. J.W.M., D.D.T., and S.B.L. wrote the manuscript. All authors revised and approved the final manuscript.

## Competing interests

The authors declare no competing interests.
