## [Peer Review File · Communications Biology]

Reviewers' comments:

Reviewer #1 (Remarks to the Author):

This is a highly interesting study analyzing the meaning and function of the AR receptor as potential therapeutic target in DSRCT. This is of great interest for DSRCT researchers.

Besides a new method for creating DSRCT pdx-models was successfully established.

One might argue which of these two points is the greater achievement.

The study is not only a highly valuable contribution to enable further research in DSRCT but contributes to the understanding of the disease.

From the clinical point of view, I would like to make some comments and ask some questions.

1. Obviously, there is no difference of DSRCT behavior in male and female mice. Please state if there is a difference in clinical presentation and in prognosis described in literature between affected males and female DSRCT patients – or a difference in chemotherapy sensitivity between male and female DSRCT patients? Does this support your findings?

2. Line 366: You discuss that the clinically observed beneficial effects are not due to the AR receptor. You hypothesize that the glucocorticoid receptor may play a critical role. Do you think that the accompanying inflammatory processes described in Scheer et al. 2019, DOI: 10.1002/cam4.1940 might also be related to the glucocorticoid receptor?

3. line 345: The major challenge in DSRCT treatment is not primarily “high frequency of recurrence and metastasis”, indeed no sarcoma can be cured if sufficient local therapy is not possible. The particular problem of DSRCT lies in the specific pattern of spread which makes a local therapy impossible and therefore it comes to a recurrence.

For future DSRCT research: DSRCT cells do not organize themselves into a single node. (And the patients where DSRCT occurs as a single nodule, e.g. in the testis, can be cured). Have you ever tried to find out why the DSRCT cells are not able to stick together? Beside your hypotheses, could this be another brick for understanding the disease?

Reviewer #2 (Remarks to the Author):

This manuscript by Magrath et al. presents two independent contributions:

- A technical resource: they explore in great detail and quite robustly the precise number of cells and conditions that are needed to seed DSRCT cell line-derived xenografts in mice.

- The relevance of the AR receptor in DSRCT tumors (which had previously been studied in Fine et al. and Lamhamedi-Cherradi et al.), but this time stemming from the observation that female mice are capable of bearing DSRCT xenografts.

They both have merit in themselves, and my minor comments below are only intended at bringing additional clarity to some aspects:

For the Introduction section, I would like to bring attention to this review (PMID: 36202860) where DSRCTs are placed in the broader context of small round cell sarcomas, and may help the non-expert understand how they compare to other entities that mimic some of their features.

Line 102-103: is this difference significant?

Fig 1B: Tumors from BER-injected mice at 1E2-sdherent concentration show an extreme size scattering. Could the authors speculate on the reasons for this?

Reviewer #1 (Remarks to the Author):

This is a highly interesting study analyzing the meaning and function of the AR receptor as potential therapeutic target in DSRCT. This is of great interest for DSRCT researchers. Besides a new method for creating DSRCT pdx-models was successfully established. One might argue which of these two points is the greater achievement. The study is not only a highly valuable contribution to enable further research in DSRCT but contributes to the understanding of the disease.

Thank you for your thorough review of our paper. We are glad you found it to be of great interest and appreciate your suggestions!

From the clinical point of view, I would like to make some comments and ask some questions.

1. Obviously, there is no difference of DSRCT behavior in male and female mice. Please state if there is a difference in clinical presentation and in prognosis described in literature between affected males and female DSRCT patients – or a difference in chemotherapy sensitivity between male and female DSRCT patients? Does this support your findings?

The largest studying examining survival in male versus female patients using SEER data, found no difference in patient survival based on sex. This aligns with our findings of no difference in tumor growth based on mouse sex. Studies have not examined differences in clinical presentation or chemotherapy response based on sex. We have added the following (in bold) to the discussion section, describing this data in the context of our findings (lines 420-422):

“One potential alternative explanation is that androgen is necessary for the formation of the EWSR1::WT1 fusion **but not subsequent tumor growth. This explanation could also help explain the absence of a difference in survival based on patient sex (Lettieri, 2014).**”

2. Line 366: You discuss that the clinically observed beneficial effects are not due to the AR receptor. You hypothesize that the glucocorticoid receptor may play a critical role. Do you think that the accompanying inflammatory processes described in Scheer et al. 2019, DOI: 10.1002/cam4.1940 might also be related to the glucocorticoid receptor?

More research will be necessary to understand the potential importance of the glucocorticoid receptor in DSRCT and its potential inhibition by enzalutamide. In our in vitro model, there are no immune cells so any effects on the glucocorticoid receptor must be independent of their impacts on the immune system. Therefore, we think the observations by Scheer et al. are not likely to be related to our discussion of the glucocorticoid receptor.

3. line 345: The major challenge in DSRCT treatment is not primarily “high frequency of recurrence and metastasis”, indeed no sarcoma can be cured if sufficient local therapy is not possible. The particular problem of DSRCT lies in the specific pattern of spread which makes a local therapy impossible and therefore it comes to a recurrence.

For future DSRCT research: DSRCT cells do not organize themselves into a single node. (And the patients where DSRCT occurs as a single nodule, e.g. in the testis, can be cured). Have you ever tried to find out why the DSRCT cells are not able to stick together? Beside your hypotheses, could this be another brick for understanding the disease?

Thank you for the question. It is an extremely important point and we are not aware of any research examining the reason for DSRCT’s appearance as tens to hundreds of nodules. The expression or lack of expression of certain surface proteins could be an important area for future investigation and will be a point of discussion within our lab. To further emphasize this problem in the discussion section, we have added the following sentence (lines 353-356):

“The presentation of DSRCT with tens to hundreds of independent nodules makes complete surgical resection extremely challenging and the prioritization of systemic therapy of paramount importance.”

Reviewer #2 (Remarks to the Author):

This manuscript by Magrath et al. presents two independent contributions:

- A technical resource: they explore in great detail and quite robustly the precise number of cells and conditions that are needed to seed DSRCT cell line-derived xenografts in mice.
- The relevance of the AR receptor in DSRCT tumors (which had previously been studied in Fine et al. and Lamhamedi-Cherradi et al.), but this time stemming from the observation that female mice are capable of bearing DSRCT xenografts.

They both have merit in themselves, and my minor comments below are only intended at bringing additional clarity to some aspects:

For the Introduction section, I would like to bring attention to this review (PMID: 36202860) where DSRCTs are placed in the broader context of small round cell sarcomas, and may help the non-expert understand how they compare to other entities that mimic some of their features.

Thank you for your thorough review of our manuscript and your recommendation of this review paper. We enjoyed reading it and agree it provides additional context that may be helpful for readers of our article. We have added a sentence to the introduction discussing DSRCT's inclusion as a small round cell sarcoma and referenced this review. The new sentence reads as follows (lines 39-41):

“DSRCT is a member of the small round cell sarcoma tumor type which includes other oncogenic fusion protein driven tumors such as Ewing sarcoma, CIC::DUX4, and BCOR::CCNB33.”

Line 102-103: is this difference significant?

The growth difference between JN-DSRCT-1 cells seeded from sphere or adherent culture with 1E5 cell was significant ($p=0.0004$). We have added this to the text (line 105).

Fig 1B: Tumors from BER-injected mice at 1E2-adherent concentration show an extreme size scattering. Could the authors speculate on the reasons for this?

As the tumors derived from 1E2 cells were grown for the longest time period, (over 150 days for BER tumors), small differences in cell seeding or growth can be compounded over time. We believe this is the likely reason for the larger tumor size variation seen for 1E2 tumor groups. Further, one of the 1E2 BER adherent culture tumors invaded into the peritoneum,

which is likely a better environment for DSRCT tumor growth and could explain its outsized growth relative to the other tumors. We have added the following to the text (lines 111-116):

“We observed a greater variation in tumor size in groups originating from 10^2 cells, which may be the result of small variations in cell seeding or growth coupled with a longer growth period (over 100 days). The third replicate in the BER-DSRCT 10^2 adherent group was the only tumor found to invade the peritoneum and was by far the largest tumor in its group. As the peritoneum is the natural location of DSRCT tumors, this observation may reflect the influence of the microenvironment on tumor growth.”